# Serum Response Factor-Regulated IDO1/Kyn-Ahr Pathway Promotes Tumorigenesis of Oral Squamous Cell Carcinoma

**DOI:** 10.3390/cancers15041319

**Published:** 2023-02-19

**Authors:** Mingyan Xu, Feixiang Zhu, Qi Yin, Hao Yin, Shaobin Fang, Gongwei Luo, Jie Huang, Wenxia Huang, Fan Liu, Ming Zhong, Xiaoling Deng

**Affiliations:** 1Department of Implantology, Stomatological Hospital of Xiamen Medical College & Xiamen Key Laboratory of Stomatological Disease Diagnosis and Treatment, Xiamen 361008, China; 2Department of Basic Medical Science, School of Medicine, Xiamen University, Xiamen 361104, China; 3Department of Stomatology of Second Affiliated Hospital of Shantou University Medical College, Shantou 515041, China; 4Department of Stomatology, Xiang’an Hospital of Xiamen University, Xiamen 361104, China

**Keywords:** serum response factor, oral squamous cell carcinoma, IDO1/Kyn-AhR signaling pathway, epithelial to mesenchymal transition, tumorigenesis

## Abstract

**Simple Summary:**

Oral squamous cell carcinoma (OSCC) is the most widespread malignancy of the head and neck and is characterized by a high potential for local invasion and lymph node metastasis. Serum response factor (SRF) regulates pro-carcinogenic genes in various cancers, but its role in OSCC remains unclear. The present study is the first to elucidate the role of SRF in OSCC development to our knowledge. We revealed that SRF is overexpressed in patients with OSCC, which is correlated with the depth of invasion and lymph node metastasis. Tumorigenicity assays in nude mice further proved that SRF promoted OSCC tumorigenesis in vivo. In addition, overexpressed SRF regulated the novel IDO1/Kyn-AhR signaling pathway to enhance OSCC cell migration and invasion by modulating EMT. In conclusion, we revealed a novel molecular mechanism by which SRF modulates OSCC metastasis. Our study could provide potential targets or biomarkers for OSCC diagnosis and treatment.

**Abstract:**

Serum response factor (SRF) regulates pro-carcinogenic genes in various cancers, but its role in oral squamous cell carcinoma (OSCC) remains unclear. SRF expression in 70 OSCC samples was detected via immunohistochemistry. Abundant SRF expressed in OSCC tissues was closely associated with tumor metastasis. SRF-overexpressing OSCC cells were constructed to evaluate how SRF affects OSCC cell tumorigenesis and epithelial-to-mesenchymal transition (EMT) in vitro and in vivo. Overexpressed SRF increased OSCC cell migration and invasion in vitro and tumor growth and invasion in vivo. This promoted EMT, characterized by decreased and increased expression of E- and N-cadherin, respectively. Furthermore, an analysis of RNA sequences of transcriptional targets of SRF showed that SRF transactivated the indoleamine 2, 3-dioxygenase 1 (IDO1)/kynurenine-aryl hydrocarbon receptor (Kyn-AhR) signaling pathway in OSCC cell lines. Direct SRF binding to the *IDO1* gene promoter upregulated transcription, which was detected through chromatin immunoprecipitation and dual luciferase reporter assays. Inhibiting IDO1 or AhR impaired SRF-induced migration and invasion and prevented EMT in OSCC cells. Our results demonstrated that SRF is a critical regulator of the IDO1/Kyn-AhR signaling pathway. This in turn increases OSCC cell migration and invasion by modulating EMT, which, consequently, favors OSCC cell growth and metastasis. We revealed a novel molecular mechanism through which SRF modulates OSCC metastasis. This should provide potential targets or biomarkers for OSCC diagnosis and treatment.

## 1. Introduction

Oral squamous cell carcinoma (OSCC) is the most widespread malignancy of the head and neck and is characterized by a high potential for local invasion and lymph node metastasis [1]. More than 377,000 new OSCC cases were diagnosed, and 177,000 deaths occurred globally in 2020 [2]. Although early-stage OSCC is treatable in >80% of patients, advanced OSCC has a poor survival rate, resulting in death in >70% of patients [3]. The diagnosis and treatment of OSCC have progressed over the past few decades, but the 5-year survival rate remains disappointing due to a high recurrence rate, frequent metastases, and most patients presenting with advanced disease at the time of diagnosis [4,5,6]. Therefore, the molecular mechanisms of OSCC tumorigenesis urgently need to be understood, and the key factors influencing OSCC metastasis need to be identified.

Metastasis occurs when malignant OSCC cells invade other tissues by migrating from a primary tumor site due to epithelial-to-mesenchymal transition (EMT), which is a pivotal driving mechanism [7]. The hallmark of EMT is E-cadherin downregulation followed by N-cadherin upregulation [8], which manifests as changes in specific cell morphology, loss of intercellular adhesion, and increased cell motility [9]. Aberrant expression of the transcription factors Snail [10], Twist [11], ZEB1 [12], and FOXO1 [13] can contribute to metastasis through EMT in OSCC. Serum response factor (SRF) is a transcription factor belonging to the *Mcm1*, *Agamous*, *Deficiens*, and *SRF* (MADS)-box family. It regulates several genes by binding to the DNA cis-element CC[AT]_6_GG (CArG) box and is involved in cell proliferation, migration, differentiation, apoptosis, and angiogenesis [14]. SRF is ubiquitously expressed in various cell types, where it plays a crucial role in physiological and pathological processes [15]. Emerging evidence has indicated that SRF plays crucial roles in the progression of prostate, gastric, and cervical cancers, as well as hepatocellular carcinoma, through the migration and invasion of epithelial cancer cells [16,17,18], and it is associated with poor prognosis [19]. However, the expression and role of SRF in OSCC carcinogenesis remain unknown.

Indoleamine 2,3-dioxygenase 1 (IDO1) converts tryptophan (Trp), a key regulator of innate and adaptive immunity, into the downstream catabolic product kynurenine (Kyn) [20]. Both Trp depletion and Kyn accumulation provide a favorable microenvironment for tumor cells to escape immune surveillance [21]. Kynurenine is an endogenous aryl hydrocarbon receptor (AhR) agonist that interacts with AhR and inhibits immune cell activity and proliferation, consequently suppressing the antitumor immune response [22]. High IDO1 expression in the tumor microenvironment of OSCC correlates with poor patient prognoses [23]. The signaling pathways associated with IDO1 catalytic activity play critical roles in tumor development, metastasis, and prognosis. However, the regulation of IDO1-mediated Trp-Kyn-AhR signaling in OSCC development, as well as its association with SRF function, remains unclear.

The present study found that upregulated SRF in human OSCC tissues was closely associated with the clinical features and prognosis of OSCC in vitro and in vivo. We also showed that IDO1 is a novel downstream target of SRF, which promotes EMT in OSCC cells by activating the Kyn-AhR signaling pathway and facilitates OSCC cell migration and invasion. These findings offer potential targets that should facilitate the early diagnosis of OSCC and help to develop new strategies for treating OSCC.

## 2. Materials and Methods

### 2.1. Tumor Samples

We collected OSCC and normal oral mucosa (NOM) tissues at the Affiliated Stomatological Hospital of Xiamen Medical College (Xiamen, China) between 2013 and 2020. The NOM tissues were derived from healthy individuals who had undergone orthodontic tooth extraction. Seventy OSCC tissue samples were derived from patients who had not received any medication before surgery. All OSCC specimens were stained with hematoxylin–eosin and histologically confirmed according to the criteria of the WHO Classification of Head and Neck Tumors [24]. All OSCC tissue samples were accompanied by information about the patients, such as sex, age, depth of invasion, tumor differentiation, and lymph node metastasis. Table 1 shows the clinicopathological features of the patients.

### 2.2. Cell Culture and Reagents

The human OSCC cell lines HSC3 and SAS (Collection of Research Bioresources Cell Bank, Osaka, Japan) were incubated in high-glucose Dulbecco’s modified Eagle’s medium (DMEM; Gibco BRL, Waltham, MA, USA) containing 10% fetal bovine serum (FBS; Gibco BRL, New York, NY, USA), 0.1 mg/mL streptomycin, and 100 U/mL penicillin in a 5% CO_2_ atmosphere at 37°C. The IDO1 and AhR inhibitors, palmatine chloride (PC; HY-N0110 purity > 98%) and PDM2 (HY-112629) were both purchased from MedChemExpress (Shanghai, China).

### 2.3. Immunohistochemical Staining

Tissues were fixed with 4% paraformaldehyde, embedded in paraffin, cut into 5 μm-thick sections, deparaffinized in xylene, and rehydrated with a graded ethanol series (100%, 90%, 70%, and 50%). Endogenous peroxidases were blocked with 3% H_2_O_2_, followed by antigen retrieval in boiling 10 mM trisodium citrate buffer (pH 6.0). Sections were stained using MaxVision DAB Kits (Maxim Biotechnologies, Fuzhou, China). Primary antibodies against SRF (sc-335, Santa Cruz Biotechnology, Dallas, TX, USA), E-cadherin (ab76055; Abcam, Cambridge, UK), and N-cadherin (13769-1-AP, Proteintech, Wuhan, Hubei, China) were applied. Ten non-repetitive and non-overlapping visual fields were randomly selected from each sample using the cell counting function of Image-Pro ^®^Plus (IPP) software. Ratios (%) of SRF-positive cells were calculated by counting the total number of cells in each field and number of SRF-positive cells. Samples with a mean ratio ≥10% were defined as SRF-positive. The staining intensity of SRF was quantified in five random fields per section.

### 2.4. Creation of Cell Lines Stably Overexpressing SRF

The cDNA encoding SRF was cloned into the pCDH-CMV-MCS-EF1-GFP-PURo vector (System Biosciences LLC., Palo Alto, CA, USA) to construct plasmids overexpressing SRF. These plasmids were mixed with pGag/Pol, pRev, and pVSV-G and transfected into 293T cells using Lipofectamine 2000 (Invitrogen Life Technologies, Waltham, MA, USA). Supernatants containing lentiviruses were collected from the 293 T cell medium. The SAS or HSC3 cells were transfected with supernatants containing a non-targeting lentivirus as the negative control (NC) or a lentivirus expressing SRF using 8 μg/mL polybrene (Sigma-Aldrich Corp., St Louis, MO, USA) for 14–16 h. The infectious medium was replaced with complete medium for a further 24 h. Infected cells were selected using 2 μg/mL puromycin (Sigma-Aldrich Corp.) for 12 days. The expression of SRF in infected cells was detected using quantitative real-time PCR (qRT-PCR) and western blotting.

### 2.5. Tumorigenicity Assays in Nude Mice

Four-week-old male BALB/c nude mice (Beijing Vital River Laboratory Animal Technology Co., Ltd., Beijing, China) were handled using procedures approved by the Institutional Review Board of Xiamen Medical College (20210305012). The mice were randomly divided into SAS NC, SAS SRF, HSC-3 NC, or HSC-3 SRF groups (*n* = 6 per group). SAS NC, SAS SRF, HSC-3 NC, or HSC-3 SRF stable cells (5 × 10^5^/mouse) in 0.1 mL of PBS were injected subcutaneously into the right axillae of the mice. The SAS NC and SAS SRF groups were euthanized on day 28, and the HSC3 NC and HSC3 SRF groups were euthanized on day 40. Tumor volume was calculated as
(L × W^2^)/2
where L and W represent tumor length and width, respectively.

### 2.6. Western Blotting

Total proteins were extracted from OSCC tissues and cells using RIPA lysis buffer. Protein concentrations were determined using BCA protein assay kits (Sigma-Aldrich Corp.). Equal amounts of protein were resolved via 10% sodium dodecyl sulfate-polyacrylamide gel electrophoresis and transferred onto polyvinylidene difluoride membranes. Non-specific protein binding was blocked with non-fat milk in PBS, and then, the membranes were probed at 4 °C for 14–16 h with anti-SRF, anti-E-cadherin, anti-N-cadherin, anti-IDO1, and anti-AhR polyclonal antibodies (all from Proteintech Group Inc., Rosemont, IL, USA), as well as an anti-GAPDH primary antibody diluted 1:2000 (Goodhere Biotechnology, Hangzhou, China). The membranes were then incubated with horseradish peroxidase-conjugated secondary antibodies (Biosharp, Hefei, China) for 2 h at 25 °C. The density of immunoreactive bands was visualized using enhanced chemiluminescence (Advansta, San Jose, CA, USA), which was quantified using Image J version 1.48 software (National Institutes of Health, Bethesda, MD, USA).

### 2.7. Wound Healing Assays

We seeded SAS-NC, SRF-SAS, HSC3-NC, and SRF-HSC3 cells in six-well plates and incubated them until they reached 100% density. The surfaces of cell monolayers were scratched with the tip of a 200 μL plastic pipette and then incubated in serum-free medium for 5 h. Wound healing at 0 and 5 h was evaluated as cell migration using a light microscope (Olympus Corp., Tokyo, Japan).

### 2.8. Migration and Invasion Assays

Migration and invasion were assayed using 24-well Transwell inserts. Cells (5 × 10^5^) were starved of serum for 24 h, harvested, and then resuspended in 200 µL of serum-free DMEM containing 0.1% sterile BSA. The resuspended cells were added to the upper chamber to evaluate migration or to Matrigel-coated chambers (BD Falcon) for invasion. The lower chamber contained 600 µL of DMEM with 10% FBS as a chemoattractant. The cells were incubated at 37 °C for 36 h, fixed in 95% ethanol, and then stained with 0.1% crystal violet. The migrated cells were counted in six randomly selected microscope fields. We analyzed the influence of SRF on migration and invasion by adding cells that were transiently transfected with pCGN-SRF or an empty vector (control) to the upper chamber.

### 2.9. Sequencing RNA and Hybridization

Total RNA was extracted from cells using TRIzol (Invitrogen Life Technologies, Carlsbad, CA, USA) as described by the manufacturer. The purity, concentration, and integrity of RNA were determined using a NanoDrop spectrophotometer (Thermo Fisher Scientific Inc., Waltham, MA, USA). Oligo dT magnetic beads were specifically bound to the poly (A) tail of mRNA for purification. The first cDNA strand was synthesized using random primers and reverse transcriptase with mRNA as the template. The second strand was then synthesized, and mRNA was removed. The resulting cDNA was repaired and purified using the AMPure XP system (Beckman Coulter Inc., Brea, CA, USA). The purified product was quantified with a bioanalyzer 2100 system (Agilent Technologies Inc., Santa Clara, CA, USA). A library was created and sequenced on a NovaSeq 6000 platform (Illumina Inc., San Diego, CA, USA). The read value of each gene was compared with the original expression of the gene using HTSeq statistics v. 0.9.1 (Suzhou PANOMIX Biomedical Tech Co., Ltd., Suzhou, China), and then, expression was normalized to fragments per kilobase of exon per million mapped fragments (FPKM) values. Differentially expressed genes were analyzed using DESeq v. 1.30.0 with the following screening criteria: expression differences, multiple; |log_2_ fold-change| > 1; significance, *p* < 0.05.

### 2.10. RNA Isolation and qRT-PCR

Total RNA was extracted from cells using Universal RNA Extraction Kits (Dongsheng Biotech, Guangzhou, China) as described by the manufacturer and reverse transcribed using FastQuant RT kits (Tiangen Biotech, Beijing, China). qRT-PCR proceeded using an ABI StepOne system (ABI) and SuperReal PreMix Kit (Tiangen Biotech, Beijing, China). The relative mRNA expression of *GAPDH* was calculated using the 2^−ΔΔCt^ method. The forward and reverse (5′ → 3′) primers were as follows:

*GAPDH*, CATCACCATCTTCCAGGAG and AGGCTGTTGTCATACTTCTC;

*SRF*, CGAGATGGAGATCGGTATGGT and GGGTCTTCTTACCCGGCTTG;

*IDO1*, GGAGGACATGCTGCTCAGTT and CTGGCTTGCAGGAATCAGGA.

### 2.11. Kynurenine Measurements

We measured Kyn concentrations in medium to confirm whether it was metabolized from Trp via IDO1 activation in SAS and HSC-3 cells overexpressing SRF. The cells were transfected with SRF lentivirus for 72 h, and then, Kyn concentrations (ng/mL) were measured in culture supernatants using enzyme-linked immunosorbent assay (ELISA) kits (Ruixin Biotechnology, Quanzhou, China).

### 2.12. Dual Luciferase Reporter Assays

An SRF potential binding site at the *IDO1* promoter region was predicted using LASAGNA-Search 2.0 bioinformatics software. Sequences containing SRF wild-type (WT) or mutant (mut) binding sites for the *IDO1* promoter were cloned into the PGL-3 luciferase reporter vector (Promega Corp., Madison, WI, USA) to generate pGL3-IDO1-WT1, pGL3-IDO1-mut1, pGL3-IDO1-WT2, and pGL3-IDO1-mut2 plasmids. WT or specific mutated plasmids were transfected into cells stably overexpressing SRF. Luciferase activity assays were performed using a dual-luciferase reporter assay system as described by the manufacturer (Promega Corp, Fitchburg, WI, USA), and luciferase activity was determined using a luminometer (BioTek Instruments, Winooski, VT, USA). Relative luciferase activity was normalized to Renilla luciferase activity.

### 2.13. Chromatin Immunoprecipitation (ChIP) Assay

ChIP assays were performed using a ChIP kit (Cell Signaling Technology, Danvers, MA, USA). Cells were subjected to cross-linking with 1% formaldehyde, quenching with 0.125 mmol/L glycine, and lysis. We fragmented DNA using an ultrasonicator (Sonic Corp., Mitsuho, Japan). Cross-linked protein–DNA complexes were immunoprecipitated using anti-SRF or isotype control IgG at 4 °C for 14–16 h, washed with RIPA buffer, and unlinked. The DNA fragments of input, IgG, and IP groups were purified and analyzed through qPCR using specific SRF binding site target primer. The forward and reverse (5′ → 3′) primers were as follows: SRF site 1: TGCCTCTAAAGTGAACCACAGA and TAACTGTACCTGACTGCGGG; SRF site 2:TGCACAGAGATGCTTTTGTGG and ACAGCCAGTGACCACAGTTT.

### 2.14. Signal Transduction Assays

Serum-starved OSCC cells (SAS and HSC3) were infected with lentiviruses expressing SRF and incubated with or without the inhibitors PC (10 μM; for IDO1) or PDM2 (1 nM; for AhR). The cells were then transfected with SRF lentivirus for 48 h, and then proteins were subjected to western blotting for 72 h to assay migration and invasion.

### 2.15. Statistical Analysis

Data are presented as the means ± standard errors of the mean. Between-group differences were determined by performing unpaired Student t-tests, analysis of variance, or Chi-square tests using GraphPad Prism 8 (GraphPad Software Inc., San Diego, CA, USA). Significance was set at *p* < 0.05.

## 3. Results

### 3.1. Upregulated SRF Expression Is Correlated with Tumor Invasion and Metastasis in Human OSCC Tissues

We immunohistochemically evaluated SRF expression in 70 OSCC and 27 NOM tissues. We found that SRF was undetectable or weakly expressed in NOM but significantly elevated in OSCC tissues. Notably, SRF staining was weak in NOM cell nuclei, whereas strong SRF staining was diffusely detected in the nucleus and cytoplasm of OSCC tissue (Figure 1A,B). We further assessed the proportion of SRF-positive cells and the intensity of SRF protein staining using computer-aided morphological analysis. The proportion of SRF-positive cells was much higher in OSCC than in NOM samples (45.80% vs. 3.95%; Figure 1C). Moreover, the staining intensity of SRF protein was higher in OSCC than in NOM tissues (Figure 1D). These results showed abundant SRF expression in OSCC tissues.

We analyzed correlations between SRF expression and the main clinicopathological features of OSCC to determine the potential function of SRF in OSCC tumorigenesis. Table 1 shows a significant difference between T1/T2 and T3/T4 tumors. This indicated that SRF expression was positively correlated with the invasion depth, especially in T3 and T4 OSCC (*p* = 0.008), and was associated with lymph node metastasis (*p* = 0.005). Therefore, SRF expression was abundant in specimens from patients with OSCC lymph node metastasis. However, the age, sex, and degree of tumor differentiation did not significantly differ (Table 1). The significantly increased SRF expression in OSCC tissues was positively correlated with tumor progression and lymph node metastasis, suggesting that SRF upregulation in OSCC tissues plays a critical role in tumor invasion and metastasis.

### 3.2. Overexpression of SRF Facilitates the Migration and Invasion of OSCC Cells

We overexpressed SRF in OSCC cells to determine whether it contributes to migration and invasion.. Quantitative RT-PCR and western blotting confirmed SRF overexpression in SAS and HSC3 cells transduced plasmids containing *SRF* cDNA or not (NC) (Figure 2A,B). Wound healing assays showed that SRF overexpression increased the migration of SAS and HSC3 cells within 5 h (Figure 2C,D). We found that SAS and HSC3 cells overexpressing SRF had more migratory and invasive ability than their corresponding NC groups (Figure 2E,F).

### 3.3. Overexpression of SRF Increases EMT of OSCC Cells

EMT is an essential process underlying tumor metastasis that is characterized by the loss and gain of epithelial and mesenchymal markers, respectively [25]. We found decreased and increased expression of the epithelial marker E-cadherin and the mesenchymal protein N-cadherin, respectively, in cells overexpressing SRF (Figure 3A,B). Immunofluorescence assays further confirmed the decreased E-cadherin and enhanced N-cadherin expression in SAS and HSC3 cells overexpressing SRF (Figure 3C,D). These data suggested that SRF overexpression promotes the EMT in OSCC cells.

### 3.4. Overexpression of SRF Promotes OSCC Tumorigenesis and EMT In Vivo

We further explored the carcinogenic role of SRF in vivo by injecting cell lines overexpressing SRF into xenograft nude mice. Plasmids containing *SRF* cDNA or not (NC) lentivirus were transduced into SAS and HSC3 cells with high efficiency according to GFP bioluminescence (Appendix A). Tumors were larger in HSC3-SRF than in HSC3-NC mice (Figure 4A). The volumes and weights of tumors were significantly larger on day 40 in HSC3-SRF than in HSC3-NC tumors (Figure 4B,C). The results on day 28 were also comparable between SAS-SRF and SAS-NC tumors (Appendix A). Tumor cells infiltrated peripheral tissues, including blood vessels, nerves, and muscles in HSC3-SRF, but not in HSC3-NC mice (Figure 4D). Furthermore, the staining intensity was lower and higher for E-cadherin and N-cadherin, respectively, in tumors from HSC3-SRF mice compared with levels in those form HSC-3-NC mice (Figure 4E,F). These results suggested that SRF overexpression promotes OSCC tumorigenesis and EMT in vivo.

### 3.5. SRF Activates the IDO1/Kyn-AhR Signaling Pathway in OSCC Cells

We explored the molecular mechanisms underlying the effects of SRF on OSCC carcinogenesis using RNA-seq assays. We detected 347 and 302 downregulated and upregulated genes, respectively (Figure 5A). Among the top 10 upregulated genes, expression of *IDO1*, a critical factor in tumor progression, was the most significantly enhanced (Figure 5B). The qRT-PCR results confirmed upregulated mRNA expression in OSCC cells overexpressing SRF (Figure 5C). IDO1 converts Trp into Kyn, which interacts with AhR and exerts different biological effects [26]. Here, Kyn concentrations in supernatants and cells from the SAS and HSC3 groups overexpressing SRF were both enhanced compared to those in the respective NC group (Figure 5D). Moreover, the overexpression of SRF increased IDO1 and AhR expression (Figure 5F). Immunofluorescence assays showed that AhR was mainly located in OSCC cell nuclei, and more AhR-positive cells exhibited greater intensity among SRF-overexpressing cells (Figure 5E). These results indicated that SRF can activate the IDO1/Kyn-AhR signaling pathway in OSCC cell lines.

### 3.6. Migration and Invasion of OSCC Is Facilitated by SRF through IDO1 Transcriptional Upregulation

The mechanism underlying the activation of the IDO1/Kyn-AhR signaling pathway was investigated. Bioinformatic results revealed that the promoter region of the *IDO1* gene contains two SRF-binding sites. We inserted promoter sequences containing SRF WT or mutated (mut) binding sites of the *IDO1* promoter region into pGL3 vectors and co-transfected them with plasmids overexpressing SRF into OSCC cells (Figure 6A). Dual luciferase reporter assays revealed no significant differences in luciferase activities between the pGL3-IDO1-WT1 and pGL3-IDO1-mut1 groups overexpressing SRF. However, SRF transfection increased the relative luciferase activity of cells co-transfected with pGL3-IDO1-WT2, but not with pGL3-IDO1-mut2 (Figure 6B). These results indicated that SRF directly binds to site 2 sequences of the *IDO1* promoter. We verified interactions between SRF and the *IDO1* promoter region using ChIP assays. The results showed that the overexpression of SRF remarkably enhanced the binding of SRF to the site 2 motif, whereas SRF binding to the first predicted fragment in the *IDO1* promoter region was not observed (Figure 6C). These data confirmed that SRF binds to the site 2 sequence of the *IDO1* promoter and activates *IDO1* transcription.

We then explored whether IDO1 is involved in SRF-mediated OSCC metastasis via the IDO1/Kyn-AhR signaling pathway. Migration and invasion were reduced in cells overexpressing SRF and incubated with the IDO1 inhibitor PC. The downregulation of AhR caused by SRF overexpression was reversed in cells incubated with PC (Figure 7A,B). Furthermore, PC prevented the decrease in E-cadherin and increase in N-cadherin staining intensity in cells overexpressing SRF. These findings suggested that the IDO1 inhibitor also prevented SRF-mediated EMT (Figure 7C,D). Consistent with the immunofluorescence findings, the expression of AhR and E-cadherin was increased, whereas that of N-cadherin was decreased, in cells overexpressing SRF and incubated with PC (Figure 7E). These findings collectively indicated that SRF promotes EMT in OSCC cells by directly binding to the promoter region of *IDO1*, thus facilitating OSCC cell migration and invasion.

### 3.7. Aryl Hydrocarbon Receptors Are Involved in SRF-promoted OSCC Cell Migration, Invasion, and EMT

We further confirmed that AhR is the downstream target of IDO1 by incubating OSCC cells with or without *SRF* cDNA (via transfection) and PDM2, an AhR inhibitor. The results showed that PDM2 reduced the SRF-enhanced OSCC cell migration and invasion (Figure 8A,B). Furthermore, PDM2 significantly reduced the increase in AhR intensity mediated by SRF overexpression. Conversely, inhibiting AhR reversed the diminished E-cadherin and stronger N-cadherin intensity in the SRF group (Figure 8C,D). The effects of SRF on E-cadherin and N-cadherin expression were suppressed in cells incubated with PDM2 (Figure 8E). Therefore, downstream AhR expression is required for SRF-mediated migration, invasion, and EMT in OSCC cells.

## 4. Discussion

SRF is important in cellular physiology and development and plays a role in tumorigenesis [27]. However, little is known about its role in OSCC, a common type of head and neck cancer. To our knowledge, the present study is the first to elucidate the role of SRF in OSCC development. We revealed that SRF is overexpressed in patients with OSCC and that it correlates with the depth of invasion and lymph node metastasis. In addition, SRF promoted the tumorigenesis of OSCC cells in vivo. Furthermore, the overexpression of SRF regulated the novel IDO1/Kyn-AhR signaling pathway to enhance OSCC cell migration and invasion by modulating EMT.

Abundantly expressed SRF plays a pivotal role in the development of epithelial cancers, such as gastric, prostate, renal cell, esophageal squamous cell, cervical, and breast cancers [27]. However, SRF expression in the context of OSCC development has not yet been fully explored. The initial stage of cancer metastasis involves the physical translocation of cancer cells from a primary tumor to distant tissues. To translocate during the initial phase, OSCC cells require the capacity to migrate and invade. We found that the overexpression of SRF promoted OSCC cell migration and invasion. Clinical data confirmed that elevated SRF expression is positively correlated with a more aggressive tumor phenotype and lymph node metastasis. More importantly, SRF promoted the growth of xenograft tumors in nude mice, and tumors formed by OSCC cells overexpressing SRF penetrated the underlying muscles, blood vessels, and nerves. These results suggested that SRF is a pro-metastatic factor that affects cancer cell migration and invasion during OSCC development. To determine whether upregulated SRF expression has a prognostic impact on OSCC patients, survival analysis of The Cancer Genome Atlas (TCGA) dataset for head and neck squamous cell carcinoma (HNSCC) did not reveal significant differences between the SRF transcriptional expression level and survival rate (Appendix A). This lack of significant differences might be due to the heterogenicity of expression data for HNSCC in TCGA dataset, as significant differences in specific survival parameters have been reported depending on the location of the primary tumor [28].

Transcriptional effect of SRF on downstream gene expression depends on its interaction with diverse cofactors to form a functional SRF/cofactor complex and control gene expression [15]. Despite different stimuli and cell-specific environments, myocardin-related transcription factors (MRTFs) and members of the ternary complex factor family are the main well-reported SRF cofactors [15]. As MRTFs were found to be involved in promoting tumor cell invasion and metastasis in epithelial cancers, such as pancreatic cancer, it is reasonable to speculate that the MRTF/SRF signaling pathway contributes to the upregulation of SRF expression during the neoplastic process of OSCC, which originates from the epithelium [29]. Further definitive evidence to demonstrate the role of the MRTF/SRF signaling pathway in OSCC is needed in a future study.

EMT is a crucial process of tumor cell invasion and metastasis. Decreased E-cadherin expression, along with increased levels of mesenchymal-specific proteins, such as N-cadherin, have served as markers of EMT [30]. In the present study, we found that the overexpression of SRF in OSCC cells significantly downregulates and upregulates the expression of E-cadherin and N-cadherin, respectively. In addition, the expression of E-cadherin and N-cadherin was significantly decreased and increased, respectively, in tumor-bearing mice overexpressing SRF compared with levels in controls. Changes in cell–cell adhesion factors (E-cadherin and N-cadherin) would render OSCC cells motile and enhance their migration and invasion ability. The increase in motility mediated by the overexpression of SRF might be due, at least in part, to EMT, resulting in OSCC metastasis. This is consistent with previous findings indicating that SRF promotes EMT during the development of hepatocellular [31] and colorectal [32] carcinoma as well as human gastric [33] and cervical [19] cancers. We discovered that SRF activation modulates EMT in OSCC cells to promote cell migration and invasion.

The downstream transcriptional targets of SRF should be identified to determine how it causes EMT in OSCC cells. We sequenced RNA and identified a downstream target gene in cells overexpressing SRF and a positive correlation between SRF and IDO1 expression. Moreover, bioinformatic analysis identified two potential SRF-binding sites located at the *IDO1* gene promoter. We showed that SRF binds to a site in and activates the *IDO1* promoter, thus upregulating IDO1 expression in OSCC cells. These data suggested that the SRF/IDO1 regulatory axis is involved in OSCC development. Moreover, to our knowledge, our novel findings represent the first evidence supporting the notion that SRF functions by regulating *IDO1* gene expression. To confirm the expression level of the SRF-IDO1-Kyn-AhR regulatory axis in a public database, TCGA, an open-access database, was used. The transcriptional expression of *SRF* was positively correlated with AhR expression in HNSCC, whereas it was not positively correlated with IDO1 expression (Appendix A). The lack of a correlation might also be due to the heterogenicity of expression data for HNSCC in TCGA dataset [28].

Because it plays an important role in the ability of cancer cells to evade attack by activated cytotoxic T-lymphocytes, IDO1 has been studied in many types of cancer [34]. The expression and activity of IDO1 are increased in many cancers, and this is considered a negative prognostic indicator for esophageal [35] and colorectal [36] cancers, thyroid carcinoma [37], non-small cell lung cancer [38], and OSCC [23,39]. Additionally, the IDO1-Kyn-AhR pathway is a critical regulator of the immunosuppressive microenvironment and favors the acquisition of a mesenchymal phenotype in thyroid cancer [40]. We propose that SRF promotes EMT by transactivating the IDO1-Kyn-AhR pathway. This notion was supported by our finding that SRF upregulated IDO1 expression, which consequently activated the downstream Kyn-AhR signaling pathway, in OSCC cell lines. This was accompanied by decreased E-cadherin and increased N-cadherin expression, indicating EMT activation. Furthermore, inhibiting IDO1 or AhR in OSCC cells expressing SRF abolished cell migration and invasion and prevented EMT. These results showed that SRF facilitates EMT in OSCC cells through the IDO1/Kyn-AhR signaling pathway. The EMT process is promoted by various transcription factors that induce EMT and molecular pathways [9]. The present findings are evidence of a novel mechanism through which EMT is regulated by IDO1, which consequently mediates an immunosuppressive pathway in OSCC cells. In summary, abundantly expressed SRF in OSCC tissues transcriptionally regulates the IDO1/ Kyn-AhR signaling pathway to promote the EMT process and increase OSCC cell migration and invasion.

## 5. Conclusions

This study was limited by its retrospective design, small sample size, and restricted access to clinical information. This prevented us from exploring the association between SRF expression and survival using Kaplan–Meier curves. The relationship between SRF expression and some important OSCC etiological factors, such as tobacco and alcohol consumption, require further investigation. The therapeutic potential and clinical prognostic capacity of the SRF/IDO1-Kyn-AhR signaling pathway in human OSCC also need further investigation in prospective trials. Nevertheless, the present findings showed that SRF promotes OSCC cell migration and invasion by transactivating the IDO1-Kyn-AhR signaling pathway, which increases EMT and favors OSCC metastasis. This novel SRF/IDO1-Kyn-AhR signaling pathway might provide new insights into the development of therapeutic targets or biomarkers for OSCC progression.

## Figures and Tables

**Figure 1 cancers-15-01319-f001:**
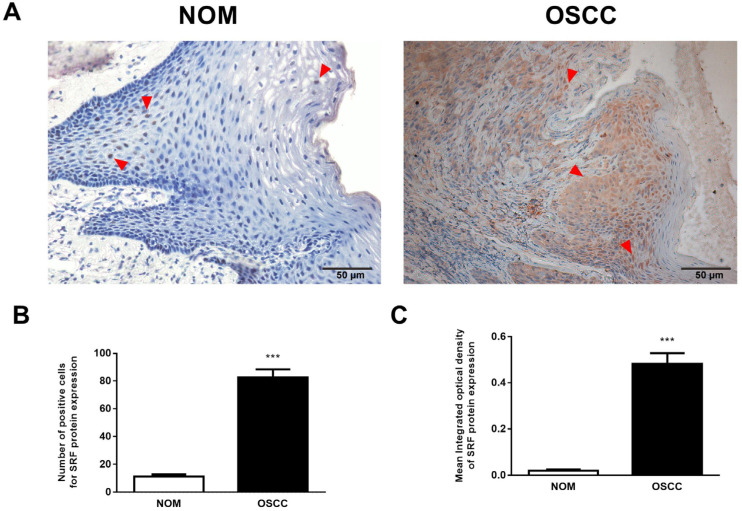
SRF is upregulated in human OSCC tissues. (**A**,**B**) Representative immunohistochemical images of SRF expression in NOM (**A**) and OSCC (**B**) tissues. representative staining of SRF marked by read triangle. Statistical analysis of proportions of SRF (**C**) positive cells and (**D**) staining intensity in OSCC and NOM tissues. Scale bar, 50 μm. *** *p* < 0.001 (Student t-tests vs. NOM). Scale bar: 50 μm. NOM, normal oral mucosa; OSCC, oral squamous cell carcinoma; SRF, serum response factor.

**Figure 2 cancers-15-01319-f002:**
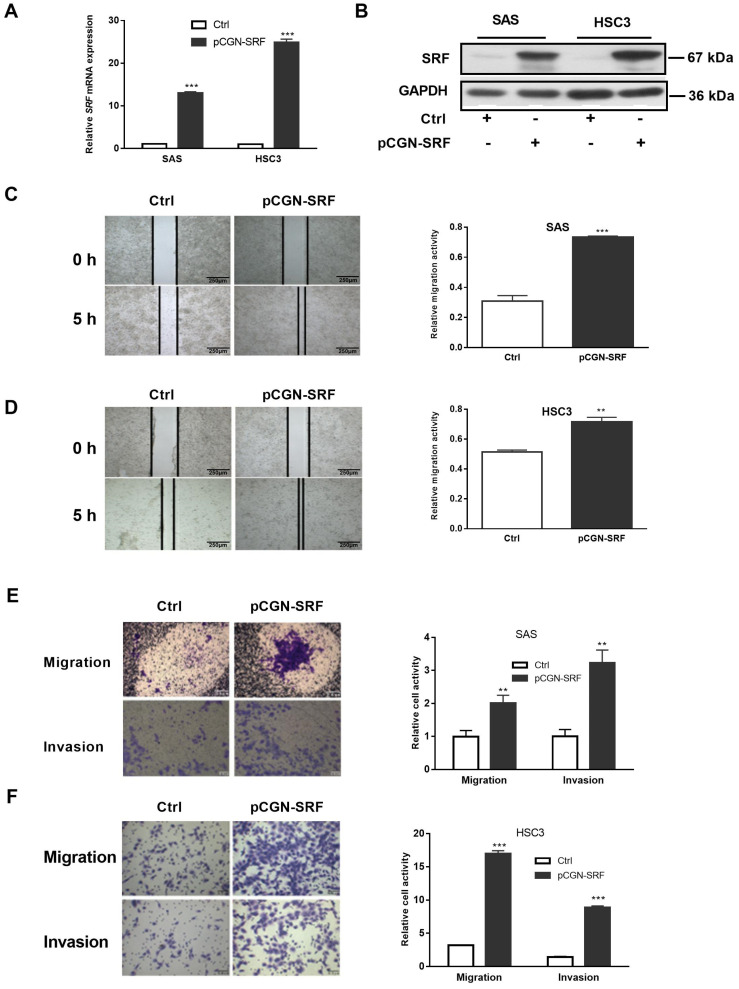
Overexpression of SRF facilitates OSCC cell migration and invasion. Serum-starved OSCC cells (SAS and HSC3) were transfected with pCGN-SRF or control vectors. The expression of *SRF* mRNA and protein were detected by performing qRT-PCR (**A**) and western blotting (**B**), respectively, at 36 h after transfection. The loading control was GAPDH. Original western blots are shown in Appendix A. Wound healing scratch assays of the migration capacity of SAS (**C**) and HSC3 (**D**) cells transfected with pCGN-SRF or control vectors. Surfaces of confluent cells were scratched 12 h after transfection. Left, representative images at 0 and 5 h after scratching. Right, quantitation of cell migration capability. Scale bar: 250μm. Migration and invasion capacity of SAS (**E**) and HSC3 (**F**) cells transfected with pCGN-SRF or control vectors using Transwells. Left, representative images; scale bar: 0.5 μm. Right, quantitation of cell migration and invasion capability. Data are representative of three independent experiments. ** *p* < 0.01, *** *p* < 0.001 vs. control. Ctrl, control; OSCC, oral squamous cell carcinoma; qRT-PCR, quantitative reverse transcription-polymerase chain reaction; SRF, serum response factor.

**Figure 3 cancers-15-01319-f003:**
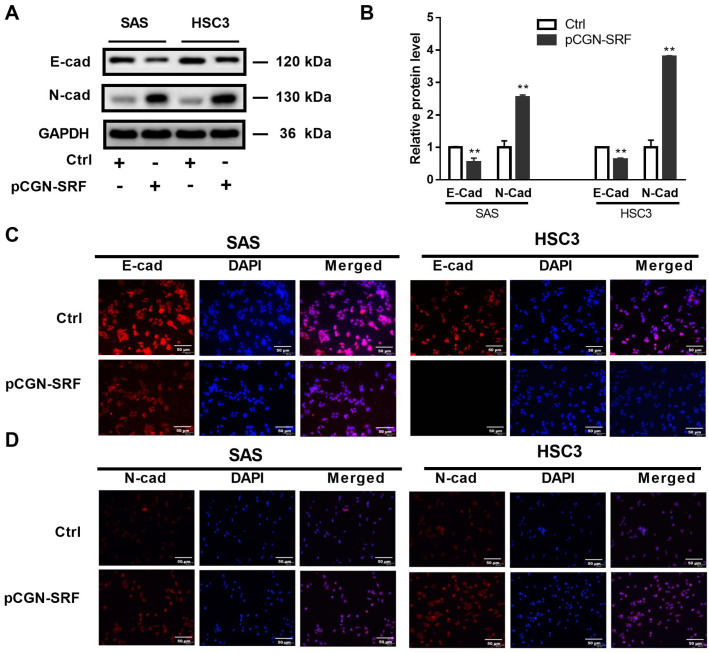
Serum response factor promotes OSCC cell epithelial-to-mesenchymal transition. Serum-starved OSCC cells (SAS and HSC3) were transfected with pCGN-SRF or Ctrl vectors. At 72 h post-transfection, the protein levels of E-cadherin and N-cadherin were detected via western blotting. GAPDH served as a protein loading control (**A**). Original western blots are shown in Appendix A. The relative band intensity was determined through densitometric analysis (**B**). Data are presented as the mean ± SD of three independent experiments. ** *p* < 0.01 vs. control vector. (**C**) Immunofluorescence staining of E-cadherin in SAS and HSC3 cells transfected with pCGN-SRF or Ctrl vectors. (**D**) Immunofluorescence staining of N-cadherin in SAS and HSC3 cells transfected with pCGN-SRF or control (Ctrl) vector. Scale bar: 50μm. Data are representative of three independent experiments. Ctrl, control; E-cad, E-cadherin; N-cad, N-cadherin; OSCC, oral squamous cell carcinoma.

**Figure 4 cancers-15-01319-f004:**
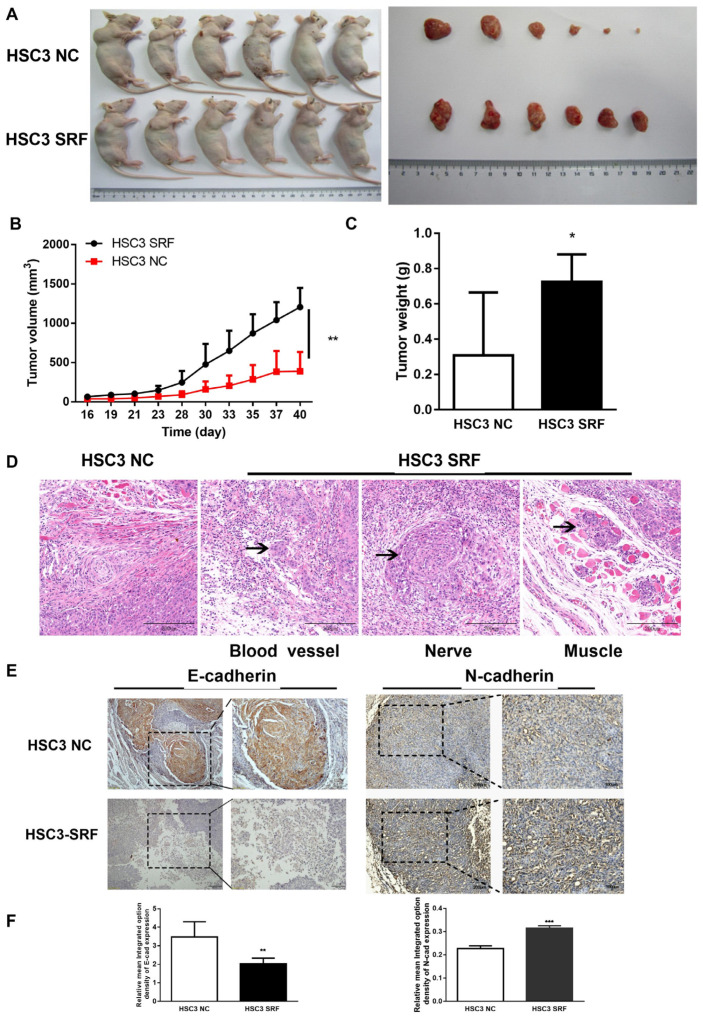
SRF overexpression promotes OSCC tumorigenesis in vivo. Xenografted nude mice (*n* = 6/group) were injected with HSC3 cells stably overexpressing SRF. The mice were euthanized, and tumors were excised from nude mice 40 days after HSC-3 cell injection (**A**). Tumor volumes were measured using calipers (**B**), and tumors were weighed at the end of the experiment (**C**). Statistical data were derived from three independent measurements and are shown as means ± SDs. Infiltration of HSC-3-SRF tumor cells (**D**). Representative immunohistochemical images of E-cadherin and N-cadherin expression in samples from HSC-3-NC and HSC-3-SRF cells (**E**). The scale bar is shown in lower-right corner. Mean optical density of E-cadherin and N-cadherin in tumor samples from HSC-3-NC and HSC-3-SRF cells (**F**). * *p* < 0.05, ** *p* < 0.01, *** *p* < 0.001, tumor samples from HSC-3-NC cells compared with HSC-3-SRF cells. NC, negative control; OSCC, oral squamous cell carcinoma; SRF, serum response factor.

**Figure 5 cancers-15-01319-f005:**
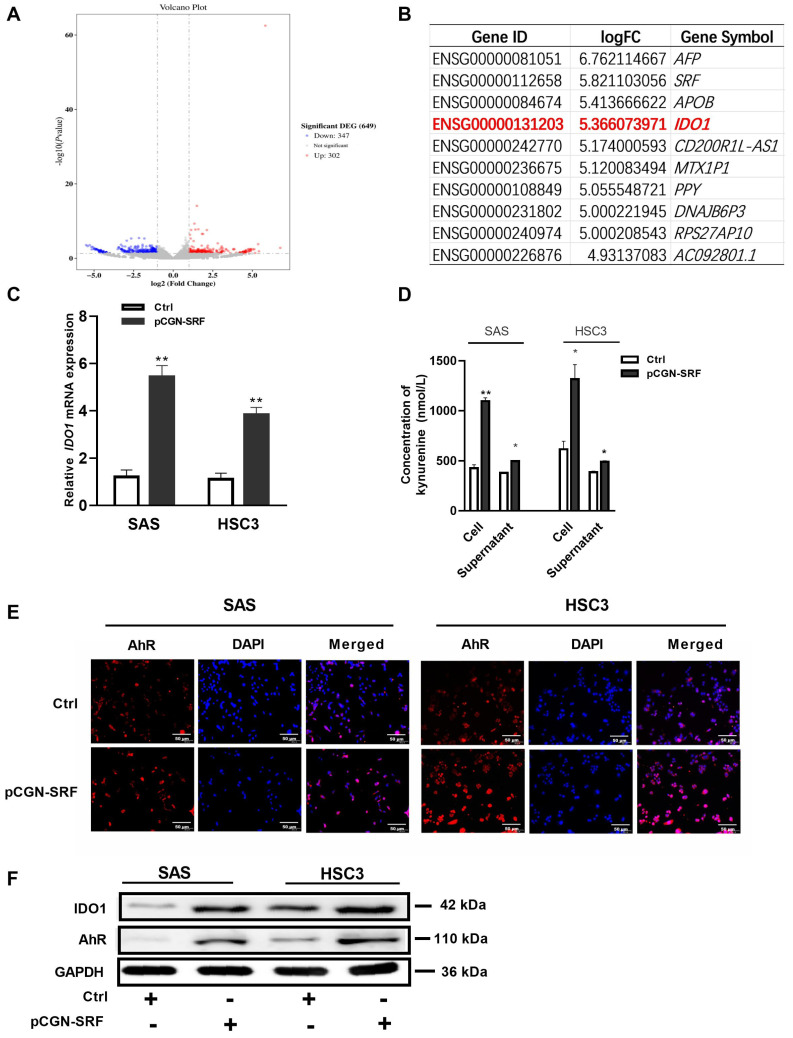
SRF activates the IDO1/Kyn-AhR signaling pathway in OSCC cells. (**A**) Volcano plots from RNAseq data of SAS cells infected with the lentivirus vector containing *SRF* cDNA. The log_2_FC is plotted on the *x*-axis and the negative log_10_FDR (adjusted *p*) is plotted on the *y*-axis. represented genes have an FDR < 0.05. Expression levels of 302 mRNAs were upregulated, whereas those of 347 mRNAs were downregulated, in stable SRF-overexpressing SAS cells compared with levels in control cells. (**B**) Top 10 genes in SAS cells upregulated by the stable overexpression of SRF. The information of *IDO1* gene expression was highlighted as red color. (**C**) qRT-PCR of *IDO1* mRNA expression in SAS and HSC3 cells transfected with pCGN-SRF or control vector for 24 h. (**D**) Kynurenine concentrations in cells transfected with pCGN-SRF for 48 h, measured using ELISA. * *p* < 0.05, ** *p* < 0.01 pCGN-SRF group compared with Ctrl group. (**E**) Immunofluorescence staining of AhR in SAS and HSC-3 cells transfected with pCGN-SRF or control vector for 48 h. (**F**) Expression of IDO1 and AhR in SAS and HSC3 cells transfected with pCGN-SRF or control vector for 48 h, detected via western blotting. The loading control was GAPDH. Original western blots are shown in Appendix A. AhR, aryl hydrocarbon receptor; Ctrl, control; DEG, differentially expressed gene; FC, fold-change; FDR, false discovery rate; IDO1, indoleamine 2,3-dioxygenase 1; Kyn, kynurenine; OSCC, oral squamous cell carcinoma; RNAseq, RNA sequencing; SRF, serum response factor.

**Figure 6 cancers-15-01319-f006:**
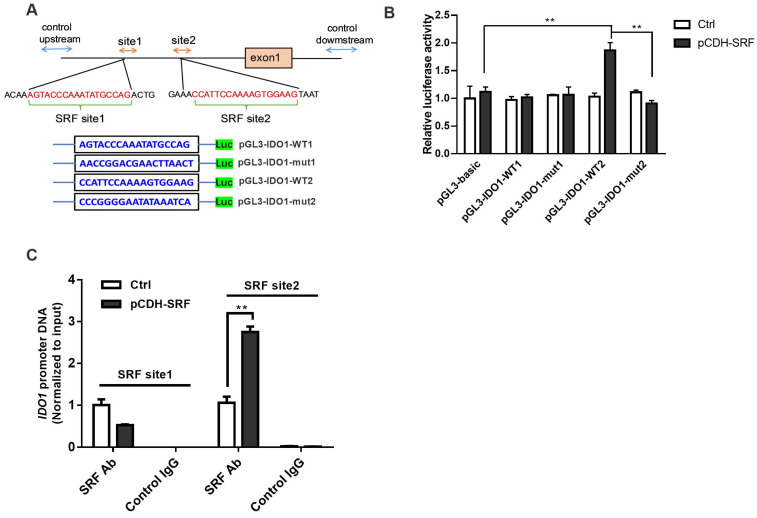
SRF is transcriptional factor that activates IDO1. (**A**) Promoter sequence of *IDO1* and sequences of WT or mut binding sites. Predicted binding sites are marked in red. (**B**) Dual luciferase reporter assay on cells co-transfected with WT plasmids or plasmids with mut-binding sites and *SRF* cDNA. (**C**) SRF binding to the *IDO1* promoter was detected using ChIP assays. ** *p* < 0.01 vs. control. ChIP, chromatin immunoprecipitation; IDO1, indoleamine 2,3-dioxygenase 1; mut, mutated; OSCC, oral squamous cell carcinoma; SRF, serum response factor; WT, wild-type.

**Figure 7 cancers-15-01319-f007:**
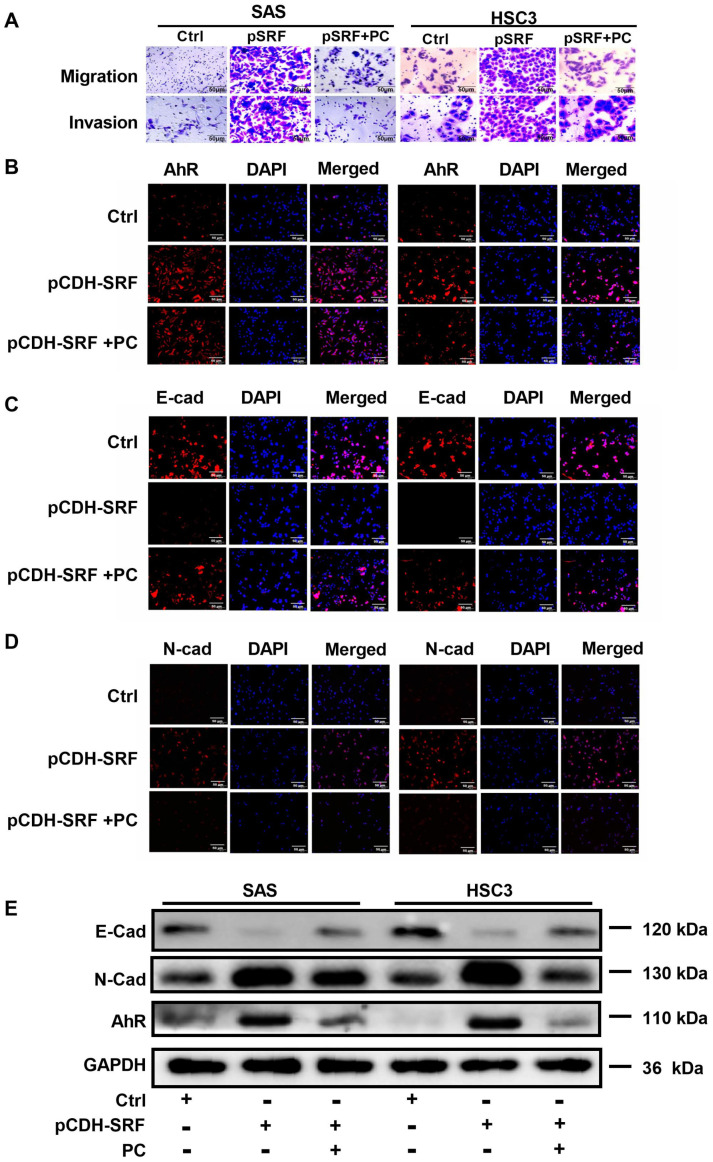
SRF facilitates the migration and invasion of OSCC by upregulating *IDO1* transcription. (**A**) Migration and invasion assays on SAS and HSC-3 cells incubated with the IDO1 inhibitor PC. Immunofluorescence staining of AhR (**B**), E-cadherin (**C**), and N-cadherin (**D**) in SAS and HSC-3 cells after adding PC. The scale bar is shown in lower-right corner. (**E**) Western blotting results of AhR, E-cadherin, and N-cadherin in SAS and HSC-3 cells after adding PC. Original western blots are shown in Appendix A. AhR, aryl hydrocarbon receptor; E-cad, E-cadherin; IDO1, indoleamine 2,3-dioxygenase 1; N-cad, N-cadherin; OSCC, oral squamous cell carcinoma; PC, palmatine chloride; SRF, serum response factor.

**Figure 8 cancers-15-01319-f008:**
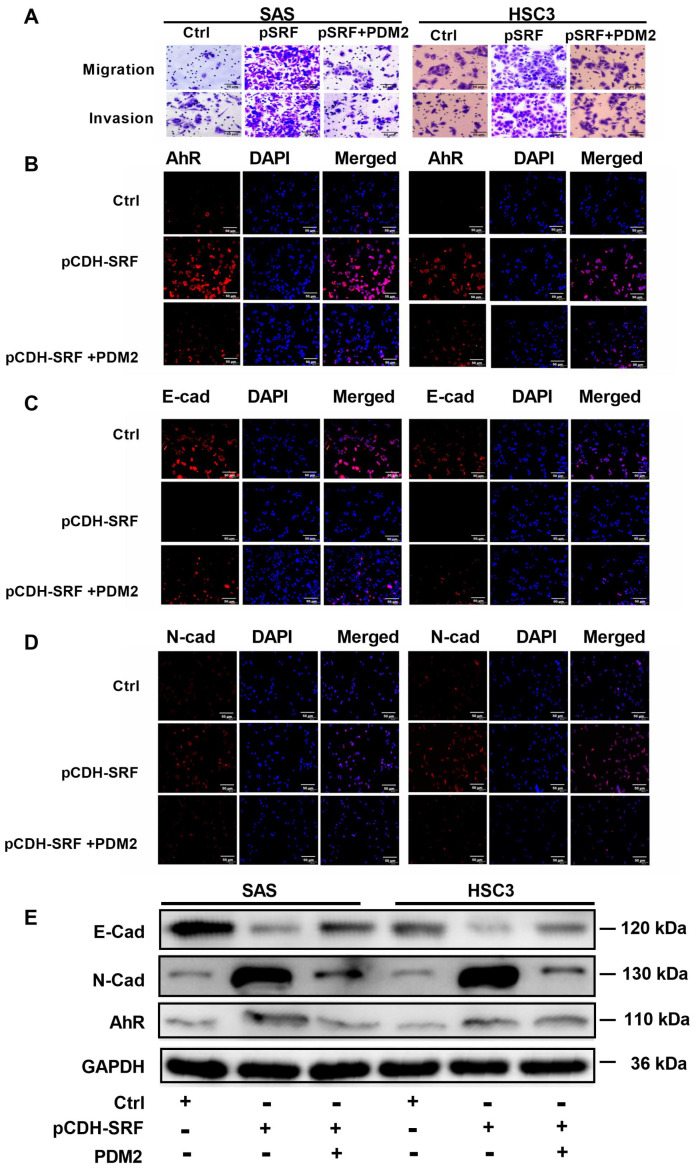
AhR is involved in the function of SRF in carcinogenesis. (**A**) Migration and invasion assays on SAS and HSC-3 cells after adding the AhR inhibitor PDM2. Immunofluorescence staining of AhR (**B**), E-cadherin (**C**), and N-cadherin (**D**) in SAS and HSC-3 cells after adding PDM2. The scale bar is shown in lower-right corner. (**E**) Western blotting results of AhR, E-cadherin, and N-cadherin in SAS and HSC-3 cells after adding PDM2. Original western blots are shown in Appendix A. AhR, aryl hydrocarbon receptor; E-cad, E-cadherin; N-cad, N-cadherin; OSCC, oral squamous cell carcinoma; PDM2, AhR inhibitor; SRF, serum response factor.

**Table 1 cancers-15-01319-t001:** Relationship between SRF expression and clinicopathological characteristics of patients with oral squamous cell carcinoma.

Clinical Parameters	SRF Expression			*p*-Value ^1^
	All Cases	High	Low	
	*n* = 70	*n* = 56	*n* = 14	
Sex				0.393
Male	42	35	7	
Female	28	21	7	
Age (y)				
<60	37	29	8	0.719
≥60	33	27	6	
Depth of invasion				0.008 *
T1/T2	51	37	14	
T3/T4	19	19	—	
Lymph node metastasis				0.005 *
Negative	42	29	13	
Positive	28	27	1	
Differentiation				0.546
Well	40	31	9	
Moderate/poor	30	25	5	

^1^ Chi-square test; * *p* < 0.05 indicates significance; SRF = serum response factor.

## Data Availability

The raw data supporting the conclusions of this article will be made available by the authors, without reservation.

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
