# Peer review of "Serum Response Factor-Regulated IDO1/Kyn-Ahr Pathway Promotes Tumorigenesis of Oral Squamous Cell Carcinoma"

_cancers, 2023, doi:10.3390/cancers15041319_

Round 1
Reviewer 1 Report
This study has demonstrated that SRF promotes EMT of OSCC by transactivating the IDO1-Kyn-AhR pathway. The roles of IDO1-Kyn-AhR axis in promoting cancer or immune pathogenesis have been pretty much known. SRF transactivates IDO-1 promoter to elicit the pathogenetic cascade being unraveled in this work is novel.
Concerns:
1. Authors should perform experiments or carry out vigorous discussion on why SRF is upregulated in OSCC or HNSCC.
2. The SRF-IDO1-Kyn-AhR regulatory axis needs to be validated using TCGA and/or GEO datasets to strengthen the findings of this study.
3. In author’s clinical cohort, the survival implications of SRF and/or downstream effectors should be assessed. Similarly, the survival analysis should be done in datasets using univariate and multivariate modes.
4. Preclinical SAS cell study is lacking.
5. CHIP would be needed to show the absence of binding of SRF to the first predicted segment in IDO promoter. Otherwise, SAM system should be imposed to further view the transactivation of endogenous IDO1 expression via the 2nd segment.
Author Response
This study has demonstrated that SRF promotes EMT of OSCC by transactivating the IDO1-Kyn-AhR pathway. The roles of IDO1-Kyn-AhR axis in promoting cancer or immune pathogenesis have been pretty much known. SRF transactivates IDO-1 promoter to elicit the pathogenetic cascade being unraveled in this work is novel.
Comments: Authors should perform experiments or carry out vigorous discussion on why SRF is upregulated in OSCC or HNSCC.
Response: Thank you very much for this comment. We agree that further discussion on the mechanism underlying the upregulation of SRF expression in OSCC is needed. The transcriptional effect of SRF on the expression of downstream genes depends on SRF interaction with diverse cofactors to form a functional SRF/cofactor complex that controls gene expression [1]. Despite different stimuli and cell-specific environments, myocardin-related transcription factors (MRTFs) and members of the ternary complex factors (TCFs) are the main well-reported SRF cofactors [1]. Since MRTFs were found to be involved in promoting tumor cell invasion and metastasis in epithelial cancers, such as pancreatic cancer, it is possible that an MRTF/SRF signaling pathway might contribute to the upregulation of SRF during the neoplastic process of OSCC, which originates from the epithelium [2]. Further definitive evidence that demonstrates the role of the MRTF/SRF signaling pathway in OSCC is needed in a future study. We revised the discussion and refined the limitations in the current version of this manuscript to address in Discussion part line465-474.
- The SRF-IDO1-Kyn-AhR regulatory axis needs to be validated using TCGA and/or GEO datasets to strengthen the findings of this study.
Response: Thank you very much for pointing this out. As suggested by the reviewer, validation of the expression level of components of the SRF-IDO1-Kyn-AhR regulatory axis in the Cancer Genome Atlas (TCGA), an open-access database, is necessary. As shown in Figure 3SA, the transcriptional expression of SRF was found to be positively correlated with AhR expression in head and neck squamous cell carcinoma (HNSCC), whereas it was not positively correlated with IDO1 expression. In the current study, we aimed to explore the molecular mechanism through which the overexpression of SRF promotes OSCC tumorigenesis. However, this lack of a correlation might be due to the heterogenicity of expression data for HNSCC in TCGA dataset, as HNSCC transcriptional expression data include primary tumors located in the oral cavity, oropharynx, and larynx, and there are significant differences in transcriptional expression depending on the location of the primary tumor [3]. We revised the discussion and refined the limitations in the current version of this manuscript to address in Discussion part (line496-501).
- In author’s clinical cohort, the survival implications of SRF and/or downstream effectors should be assessed. Similarly, the survival analysis should be done in datasets using univariate and multivariate modes.
Response: Thank you very much for this suggestion. We have discussed the survival analysis based on SRF in our clinical cohort in the Discussion section. This study was limited by its retrospective design, small sample size, and restricted access to clinical information. This prevented us from exploring the association between SRF expression and survival using Kaplan–Meier curves. In determining whether the upregulation of SRF expression has a prognostic impact on OSCC patients, survival analysis of TCGA dataset (HNSCC) did not reveal significant differences between SRF transcriptional expression levels and survival rates. This lack of significant differences might also be due to the heterogenicity of expression data for HNSCC in TCGA dataset, as it has been reported that there are significant differences in specific survival depending on the location of the primary tumor [4]. We revised the discussion and refined the limitations in the current version of this manuscript to address in Discussion part (line458-564).
- Preclinical SAS cell study is lacking.
Response: Thank you very much for this comment. We agree that preclinical SAS cell study experiments are necessary. SAS cells were used in a xenograft nude mouse model to detect the tumorigenicity of OSCC cells. The results were consistent with those of HSC3 cells, and these data are shown in Supplementary Figure 2S.
- CHIP would be needed to show the absence of binding of SRF to the first predicted segment in IDO promoter. Otherwise, SAM system should be imposed to further view the transactivation of endogenous IDO1 expression via the 2ndsegment.
Response: Thank you very much for this important suggestion. We agree that a ChIP assay to detect the binding of SRF to the first predicted segment of the IDO1 promoter is necessary. As shown in the Figure 6C, the overexpression of SRF remarkably enhanced the binding of SRF to the site 2 motif, whereas SRF binding to the first predicted fragment in the IDO1 promoter region was not observed.
- Onuh, J.O.; Qiu, H. Serum response factor-cofactor interactions and their implications in disease. FEBS J 2021, 288, 3120-3134, doi:10.1111/febs.15544.
- Song, Z.; Liu, Z.; Sun, J.; Sun, F.L.; Li, C.Z.; Sun, J.Z.; Xu, L.Y. The MRTF-A/B function as oncogenes in pancreatic cancer. Oncol Rep 2016, 35, 127-138, doi:10.3892/or.2015.4329.
- Shaikh, I.; Ansari, A.; Ayachit, G.; Gandhi, M.; Sharma, P.; Bhairappanavar, S.; Joshi, C.G.; Das, J. Differential gene expression analysis of HNSCC tumors deciphered tobacco dependent and independent molecular signatures. Oncotarget 2019, 10, 6168-6183, doi:10.18632/oncotarget.27249.
Leon, X.; Pujals, G.; Sauter, B.; Neumann, E.; Pujol, A.; Quer, M. Differential characteristics of patients with squamous cell carcinoma of the head and neck with no history of tobacco or alcohol use. Acta Otorrinolaringol Esp (Engl Ed) 2023, doi:10.1016/j.otoeng.2022.02.008
Reviewer 2 Report
Anti SFR is a nuclear immunohistochemical marker, it should appear in the nucleus on the image. Figure 1A/B shows non-specific, cytoplasmic staining of tumor cells.
Comment: a better picture should be found.
Figure 1. SRF is upregulated in human OSCC tissues
The images are labeled incorrectly!
Author Response
Anti SFR is a nuclear immunohistochemical marker, it should appear in the nucleus on the image. Figure 1A/B shows non-specific, cytoplasmic staining of tumor cells.a better picture should be found. Figure 1. SRF is upregulated in human OSCC tissues.
The images are labeled incorrectly!
Response: Thank you very much for pointing this out. In fact, we have re-examined SRF immunohistochemical results and found that SRF was undetectable or weakly expressed in the NOM group but significantly elevated in OSCC tissues. Notably, SRF staining was weak in NOM cell nuclei, whereas strong SRF staining was diffusely detected in the nucleus and cytoplasm of OSCC tissue. We have replaced the image of the SRF immunohistochemical result in Figure 1, and the label has been corrected according to the reviewer’s suggestion.